# Phage Therapy for Mycobacterium Abscessus and Strategies to Improve Outcomes

**DOI:** 10.3390/microorganisms9030596

**Published:** 2021-03-14

**Authors:** Abdolrazagh Hashemi Shahraki, Mehdi Mirsaeidi

**Affiliations:** Division of Pulmonary and Critical Care, Sleep and Allergy, Miller School of Medicine, University of Miami, Miami, FL 33101, USA; axh1558@med.miami.edu

**Keywords:** phage therapy, mycobacterial, *Mycobacterium abscessus*, mycobacteriophages

## Abstract

Members of *Mycobacterium abscessus* complex are known for causing severe, chronic infections. Members of *M. abscessus* are a new “antibiotic nightmare” as one of the most resistant organisms to chemotherapeutic agents. Treatment of these infections is challenging due to the either intrinsic or acquired resistance of the *M. abscessus* complex to the available antibiotics. Recently, successful phage therapy with a cocktail of three phages (one natural lytic phage and two engineered phages) every 12 h for at least 32 weeks has been reported against a severe case of the disseminated *M. abscessus* subsp. *massiliense* infection, which underlines the high value of phages against drug-resistant superbugs. This report also highlighted the limitations of phage therapy, such as the absence of lytic phages with a broad host-range against all strains and subspecies of the *M. abscessus* complex and also the risk of phage resistant bacteria over treatment. Cutting-edge genomic technologies have facilitated the development of engineered phages for therapeutic purposes by introducing new desirable properties, changing host-range and arming the phages with additional killing genes. Here, we review the available literature and suggest new potential solutions based on the progress in phage engineering that can help to overcome the present limitations of *M. abscessus* treatment.

## 1. Introduction

The *M. abscessus* complex is a group of rapidly growing, multidrug-resistant, nontuberculous mycobacteria (NTM) that are responsible for a wide spectrum of lung, skin, and soft tissue diseases; central nervous system and ocular infectious diseases; and bacteremia in both healthy and immunocompromised individuals [1]. The members of the *M. abscessus* complex are classified at the subspecies level to *M. abscessus* subsp. *abscessus* and *M. abscessus* subsp. *bolletii*, and *M. abscessus* subsp. *massiliense* comb. nov [2]. Similar to infections from other NTM, *M. abscessus* infections are thought to be exclusively acquired by exposure to contaminated soil or water, although human-to-human transmission of *M. abscessus* infections has been suggested in patients with cystic fibrosis [3].

The members of *M. abscessus* are a new “antibiotic nightmare” as one of the most resistant organisms to chemotherapeutic agents [4]. It has been reported that macrolide-containing regimens resulted in sputum culture conversion only in 34% and 54% of the new *M. abscessus* subsp. *abscessus* and *M. abscessus* subsp. *massiliense* patients, respectively, while in refractory disease, sputum culture conversion occurs only in 20% of patients, with no significant difference across subspecies [5]. With the currently recommended regimens, the pulmonary disease outcomes of the *M. abscessus* subsp. *abscessus* are quite similar to extensively drug-resistant tuberculosis [5] indicating the serious challenge in treating *M. abscessus* infections. *M. abscessus* is resistant to most classes of antibiotics, including macrolides, aminoglycosides, rifamycins, tetracyclines, and β-lactams [4]; as a result, there are a wide range of treatment strategies for *M. abscessus* infection using prolonged antimicrobial drug therapy, with significant side effects, and therapies often need to be changed or stopped. Generally, the treatment response rates are higher in patients with *M. abscessus* subsp. *massiliense* lung disease than *M. abscessus* subsp. *abscessus* lung disease [6] or *M. abscessus* subsp. *bolletii* [7].

*M. abscessus* also shares many virulence genes with *M. tuberculosis* [8]; however, Wee et al., reported 811 species-specific genes present in *M. abscessus,* with a high number of species-specific transcriptional regulator genes which may help in the survival of this bacterium in the environment and in the human body [9]. A successful treatment strategy should cover biofilm formation, prolonged intracellular survival, colony variant diversity, and inflammation. Successful treatments for superbugs, such as members of the *M. abscessus* complex, require new approaches beyond routine antibiotic therapy.

## 2. New Alternatives for Drug-Resistant *M. abscessus*

To combat multidrug-resistant (MDR) bacteria such as *M. abscessus*, breakthrough strategies that go beyond classical antibiotic mechanisms are urgently needed. Some of those strategies include using natural products with antimicrobial effects extracted from plants or other sources [10], nanoparticles that possess antimicrobial properties that can overcome common resistant mechanisms [11], combinations of different antibiotics as a novel treatment [12], structural alterations/modification of the existing antibiotic classes [13], antimicrobial peptides [14] pathogen-specific monoclonal antibodies [15], antibody–antibiotic conjugates [16], microbiota transplants [17], modulations of the small regulatory RNAs (sRNAs) by specific drugs [18], inhibiting the evolution of the drug resistant genes by lowering the mutation rate [19], synthetic or natural polymers [20], vaccination against superbugs [21] and therapeutic bacteriophages (Figure 1) [22].

Therapeutic bacteriophages are pathogen-specific and safe for human tissues [23]. Bacteriophages (or phages) are the most abundant organisms on Earth (10^31^ particles) and are distributed in soil, water, and air in different ecosystems and surfaces inside and outside of the human and animal body, wherever microbes can grow [24]. Phages replicate through two primary life cycles in their hosts. In the lytic cycle, sometimes referred to as virulent infection, the infecting phage ultimately kills the host cell to produce many of their own progeny. In the lysogenic cycle, sometimes referred to as temperate or nonvirulent infection, the infecting phage does not kill the host cell, instead using it as a refuge where the phage exists in a dormant state. In 2019, Dedrick et al. [25] reported the successful treatment of a 15-year-old lung-transplant patient who suffered from disseminated *M. abscessus* subsp. *massiliense* infection with mycobacteriophages with no adverse effects, suggesting that phage therapy (PT) can be a strong alternative and a practical solution for *M. abscessus* complex infection. Here, we reviewed the recent findings on phage manipulation and PT as a promising strategy to combat *M. abscessus* complex infection and further discuss the potential future direction of PT application against the *M. abscessus* complex as a model for combating other MDR bacteria. 

## 3. The History of Phage Therapy

Bacteriophages, Latin for “bacteria eaters,” were independently discovered by two microbiologists, Frederick Twort and Felix d’Herelle, in the late 1910s [26,27]. D’Herelle used phages to cure four patients who were suffering dysentery in 1917 [28] and later in 1923 to halt outbreaks of cholera in India and plague in Egypt [26] as the first applications of phages for treating a disease. In the 1930s and 1940s, bacteriophage products were commercially available in Western countries such as France, Britain, Germany, Italy, and the United States [29]. In a world before antibiotic discovery, PT was one of the many possible treatment options against infectious diseases. However, for various nation-specific reasons, PT declined in most Western countries during World War II, shortly before the triumph of penicillin [30] but persisted in the USSR, even though the Soviets had established mass antibiotic production by 1950 [29]. WHO-sponsored studies in Pakistan in the 1970s examined d’Herelle’s claims for the “therapeutic effectiveness of phage”, comparing phages with antibiotics (tetracyclines) against cholera, and showing an effectiveness equivalent to tetracycline [31,32]. PT was successfully used to treat open wound infection caused by *Staphylococci* and *Streptococci* in 6000 Soviet soldiers in the war between the Soviet Union and Finland [33]. Felix d’Herell and George Eliava (a Georgian microbiologist) established the Eliava Institute in 1923, currently active as one of the world’s top centers for bacteriophage research (https://eliavaphagetherapy.com/about-eliava-institute/, accessed on 12 March 2021).

There are many reasons for the eclipse of the PT concept after its early introduction. The primary reason was the introduction of antibiotics with many advantages, including a broad spectrum of action, easy production, and greater stability than phages. Other reasons include the detection of phage-resistant bacteria by d’Herelle [34] and others in early studies [35] and hard-to-generate reproducible results [36].

## 4. Mycobacteriophages’ Biology and Classification

Mycobacteriophages are viruses that infect mycobacterial hosts. All the characterized mycobacteriophages are double-stranded DNA (dsDNA) tailed phages belonging to the order *Caudovirales* and mostly to the family *Siphoviridae* and few to the family *Myoviridae*. *Siphoviridae* is characterized by relatively long flexible noncontractile tails, whereas *Myoviridae* contain contractile tails [37]. Currently, over 11,000 mycobacteriophages have been isolated, and 2000 of them have been sequenced [38]. Mycobacteriophages are classified in 20 clusters (A through T) and eight sequenced singletons [38] which are mostly available at https://phagesdb.org (accessed on 12 March 2021) (Figure 2). The largest is Cluster A and the smallest—aside from the eight singletons—are Clusters M, N and O, each with fewer than five phages [38]. In some clusters, such as Cluster G, the phage’s genomes are extremely similar (138 nucleotides variation between Angel and BPs) [35]; however, some of the other clusters are more diverse and can be further divided into subclusters. For example, Cluster A can be divided into at least nine subclusters [39].

Many isolated mycobacteriophages are recovered by using *M. smegmatis* mc^2^ 155 as a host [37]; however, the use of other mycobacterial species that are known to be human pathogens for phage isolation will likely give distinct landscapes of genetic diversity of mycobacteriophages. Table 1 summarizes multiple applications of mycobacteriophages reported in the literature. Mycobacteriophages can have a variety of preferences for different mycobacterial hosts. Some phages (e.g., Bxz2, D29 and L5) have broad host-ranges and can generate plaque on many species of mycobacteria, whereas others (e.g., Barnyard and Black) have very narrow preferences and only infect *M. smegmatis* [42]. It has been suggested that phages can “arrive” at a common host (*M. smegmatis* mc^2^155) by traveling from numerous phylogenetically distinct hosts as phages can expand their host-range through mutations in tail genes [39]. Some phages with a broad host-range, such as D29, are being used for different purposes such as the evaluation of drug susceptibility or PT in the genus of *mycobacterium* (Table 1). In this review, we will only focus on the therapeutic application of mycobacteriophages, particularly in relation to *M. abscessus*.

## 5. Phage Therapy against Mycobacterial Infections

### Studies Related to M. tuberculosis, M. avium, and M. ulcerans

The efficiency of PT against mycobacterial disease had been reported many years ago, when Sula et al. treated *M. tuberculosis*-infected guinea pigs with three phages; DS-6A, GR-21/T, and My-327 [55]. Almost 20 years later, *M. tuberculosis* and *M. avium* were targeted in macrophages using TM4 phage particles delivered by nonpathogenic *M. smegmatis* cells [56,57], which led to a substantial decrease in *M. tuberculosis* and *M. avium* titers in animal models. A single subcutaneous injection of the mycobacteriophage D29 showed a significant decrease in footpad pathology associated with a reduction of the *M. ulcerans* burden. Additionally, D29 treatment induced increased levels of IFN-γ and TNF in *M. ulcerans*-infected footpads, correlating with cellular infiltrates of a lymphocytic/macrophagic profile [59]. There are a limited number of mycobacteriophages that are lytic for *M. tuberculosis* [60,61] and other mycobacteria such as *M. avium* (Table 2) [42,57,61].

## 6. *M. abscessus*-Related Study: A Successful Clinical Model

Most mycobacteriophages are recovered on *M. smegmatis* as a host. Therefore, their lytic effect on clinically important mycobacteria species, such as members of the *M. abscessus* complex, has not been well studied. Recently, a 15-year-old patient with cystic fibrosis (homozygous for ΔF508) was diagnosed with disseminated *M. abscessus* subsp. *massiliense* infection and treated successfully with PT [25]. The patient was referred for a lung transplant in the UK. The patient was chronically infected with *Pseudomonas aeruginosa* and *M. abscessus* subsp. *massiliense* and was on anti-NTM treatment for 8 years before lung transplantation. Seven months after the transplant, when the patient was on immunosuppressive drugs and multiple intravenous (I.V.) antibiotics, a disseminated infection caused by *M. abscessus* subsp. *massiliense* was diagnosed. A skin infection also developed a few weeks later. Given that anti-NTM treatment could not improve *M. abscessus* subsp. *massiliense* infection, PT was considered as an alternative treatment [25]. After designing the phage cocktail (see below), the patient received a single topical test in the sternal wound and I.V. therapy with a three-phage cocktail (10^9^ plaque-forming units per dose of each phage) every 12 h for at least 32 weeks. After 9 days, the patient was discharged, and 12 h I.V. administration of the cocktail was continued. The patient continued to improve clinically with the healing of surgical wounds and skin lesions and improvement of the lung function with no side effects. Over the course of the PT, *M. abscessus* was not isolated from serum or sputum but recovered from skin lesions until 121 days after PT initiation. This is the first case report of successful PT against mycobacterial disease using either wild or engineered phages (Table 3).

To design the phage cocktail, the researchers found only mycobacteriophage Muddy to be a lytic phage against recovered *M. abscessus* subsp. *massiliense* (strain GD01) after an intensive screening of the available mycobacteriophages [25]. ZoeJ was able to lysis GD01 with reduced efficiency of plating [25]. Further analysis of ZoeJ showed that this phage has extensive sequence similarity to TM4 with broad host-ranges (able to infect fast- and slow-growing mycobacteria) similar to other members of the K2 subcluster (TM4) (Table 2) [61]. The precise deletion of a gene (repressor gene 45) resulted in an engineered phage (ZoeJΔ45) with lytic capability similar to TM4 [61]. PBs was found to infect *M. tuberculosis* with low efficiency. PBs has a detectable similarity in tail genes (genes 14–19) with those phages able to effectively infect *M. tuberculosis* such as TM4, L5, D29, Che12, and Bxz2 [64]. Frequent culturing resulted in the isolation of PBs mutants able to infect *M. tuberculosis* [64]. The culturing of PBs on GD01 also resulted in the isolation of a host-range mutant (HRM10) able to infect GD01 [25]. This mutant had a single base substitution in portal gene 3 C2083T, conferring R66W amino acid change. Moreover, gene 33 (repressor gene) was removed by phage engineering from the mutant PBs (HRM10), enabling it to effectively kill GD01 [25]. These findings suggest that novel genetic approaches can significantly increase the potential of PT as a therapeutic tool against superbugs such as *M. abscessus*. As the natural repertoire of *M. abscessus* phages is unknown since there has been no systematic isolation and characterization of the *M. abscessus* complex specific mycobacteriophages, the interaction of mycobacteriophages with *M. abscessus* complex and their evolutionary history is still undetermined.

## 7. Strategies to Improve Phage Therapy Outcome

Several approaches can be used to improve PT against MDR bacteria such as *M. abscessus*, including cocktail therapy, genomic engineering of phages, increasing the host-range of current lytic phages, and weaponizing phages with CRISPR-Cas technology.

### 7.1. Bacteriophage Cocktails

Monophage therapy involves the application of only a single phage when sufficiently wide host-range phages are available or clinically, following careful matching between pathogens and individual phage isolates [65]. Although monophage therapy has had some successes, phage cocktails (multiple phage types possessing a diversity of host-ranges) [66] are one of the best ways to increase the success of PT by reducing the risk of the development of phage resistant bacteria [67]. The different potential susceptibility of the bacterial strains to the phages can be addressed by using a phage cocktail to treat the wide range of bacterial strains/species that can cause clinical infections. None of the phages used in the recent PT study against *M. abscessus* subsp. *massiliense* was effective on other clinical strains [25], indicating that we are far from being able to formulate an effective universal phage cocktail against all strains and subspecies of pathogens such as *M. absecssus* complex. After exposing a larger bacterial culture of strain GD01 in broth culture to the designed phage cocktail, the researchers recovered some survivors resistant to the engineered phages (BPs33ΔHTH-HRM10 and ZoeJΔ45) (Table 3) [25]. The detection of this phage resistant subpopulation of strain GD01 also shed light on another limitation of PT—the rapid evolution of phage resistant bacteria—which could be solved by having multiple effective phage cocktails. The plasma half-life depends on the phage type and the biochemistry of the capsid and varies significantly among phages (60 min for phage T7 to 6 h for phage λ) [68]; thus, using different phages in a cocktail can increase the bioavailability of phages in the body and the therapeutic impact of PT. The plasma half-life of Muddy, BPs, and ZoeJΔ45 and its impact on PT outcome need further study, but a weak antibody response was detected against these phages over 220 days of treatment (every 12 h for at least 32 weeks) [25], indicating that PT is safe treatment approach even for chronic infectious disease.

### 7.2. Phage Engineering

There are many encouraging reasons for scientists in the field of PT to employ the genetic engineering approaches for phage manipulation. Phages usually have genes encoding proteins such as integrases and repressors (required for lysogeny) which restrict their therapeutic application [25,61]. Furthermore, phages carry genes with unknown functions [37], antibiotic resistance [69], or bacterial virulence factors [70] which might have downstream side effects. In addition, searching for and finding natural lytic phages for different pathogenic strains and species is costly and time-consuming; thus, genetic engineering approaches could be useful to modify the available phages that already exist in phage banks, such as the mycobacteriophages bank: https://phagesdb.org/phages/Gabriela/ (accessed on 12 March 2021) [37]. For example, to find a suitable lytic phage for *M. abscessus* subsp. *massiliense*, >10,000 isolated mycobacteriophages were screened, a process which found only Muddy to be a natural killer phage and two others (ZoeJ and BPs) able to infect the isolated *M. abscessus* subsp. *massiliense* strain [25]. ZoeJ and PBs have a lysogenic life cycle, as they both express repressor genes which control their lytic cycle [61]. The repressor gene was removed from ZoeJ and PBs using a phage engineering approach to switch their life cycle from lysogeny to lytic, which allowed them to be used as lytic phages in the designed phage cocktail [25].

Many phages have a narrow host-range, which could be a significant barrier to the application of PT to treating infections caused by different strains of a single pathogen. Muddy and engineered ZoeJ and PBs did not effectively kill other clinical strains of *M. abscessus* subsp. *massiliense* (Table 3) [25], indicating that host-range is a serious challenge even at the strain level. Other findings also support that in nature many phages have a very narrow host-range able to infect only one strain of a species (strain-specific), while other phages can infect different strains of a species (species-specific) [71], highlighting the presence of an extremely complex web of phage–host interactions even within the population of a single species. Molecular biology techniques have increased our understanding of the structures of naïve phage genomes and components, and the interactions between phages and their host bacteria, which consequently is helpful in developing the genetically engineered phages able to overcome many of these limitations [72]. Methods for the genomic engineering of phages are discussed in detail elsewhere [72]; here, we briefly discuss some of the genetic engineering approaches that might be applicable to develop more effective and broad host-range *M. abscessus* specific phages.

#### 7.2.1. Bacteriophage Recombineering of Electroporated DNA (BRED)

BRED was first described as a method for engineering phage genomes of mycobacteriophage Che9c by coelectroporesis of purified phage DNA and dsDNA, recombineering substrates into host cells (*M. smegmatis*) [73]. The host cell carries a plasmid that encodes proteins promoting high levels of homologous recombination, such as the RecE/RecT-like proteins, which lead to recombination between their homologous regions and the generation of recombinant phage particles [73]. BRED is a rapid and effective tool to knock out undesirable genes [25] or to study the functional features of phage genes (Figure 3) [74]. A repressor gene was successfully removed using BRED from mycobacteriophage ZoeJ and PBs in a phage cocktail designed to treat *M. abscessus* subsp. *massiliense* [25]. This method is mostly used to delete a target gene; however, it can be used for further phage genome manipulations, such as base substitutions, precise gene replacements, and the addition of gene tags [25,73]. BRED was also applied to knock out genes such as *cI* from the SPN9CC phage (*Salmonella*-targeting phage) [75], indicating that BRED can be optimized to manipulate phages of different bacterial hosts. However, the BRED system must be optimized for each bacterial species and is difficult to use in those species which are resistant to electroporation, and requires the screening of many lysis plaques as the efficiency of the engineering is 10% to 15% [73].

#### 7.2.2. Phage Engineering Using the CRISPR-Cas System

CRISPR in combination with *cas* genes is an adaptive immune system in bacteria and archaea, protecting microbial cells from invading foreign DNA such as phages [76]. CRISPR-Cas systems are currently classified into six types and further grouped into two broad classes (Class 1 or 2) based on phylogeny and activity mechanisms. Class 1 systems (types I, III, and IV) employ effector complexes containing multiple Cas proteins, while class 2 systems (types II, V, and VI) employ effector complexes containing a single Cas protein to cleave the target DNA [77]. They are characterized by distinct sets of *cas* genes with three steps of action; CRISPR adaptation, RNA biogenesis, and CRISPR-Cas interference [77]. CRISPR-Cas systems have been detected in *M. avium, M. bovis, M. tuberculosis,* and other mycobacteria but not in *M. absecssus* [78].

In the laboratory, the CRISPR-Cas12a system has been used as an effective genome editing tool in *M. smegmatis* [79]. The CRISPR interference (CRISPRi) approach was also efficiently used to repress the expression of target genes in the *M. tuberculosis* complex [80], highlighting the potential application of CRISPR-Cas systems for mycobacteriophage engineering. Moreover, BRED was unsuccessful in the recovery of Omega engineered phage [81] probably since capsid-enclosed proteins maybe are required for recircularization of the phage DNA, indicating that BRED approach might not suitable for all mycobacteriophages. To the best of our knowledge, there is no report yet regarding the application of CRISPR-Cas systems for mycobacteriophage engineering. Here we discuss the potential application of CRISPR-Cas systems in mycobacteriophage manipulation for PT.

##### CRISPR–Cas3 (Type I)

CRISPR-Cas system type I has been used to delete a nonessential gene of the T7 phage genome in *Escherichia coli* as host [82]. In the first step, wild-type phages are grown in the presence of a plasmid encoding a 120 bp of the sequences flanking the target gene (60 bp from each side) which can result in homologous recombination between the plasmid and a small proportion of the progeny phages. The progeny phages are the mixture of phages lacking the target gene, along with other wild-type phages that did not undergo homologous recombination. In the second step, the CRISPR–Cas3 system can be used to selectively remove the wild-type phages by expressing *cascade*, *cas3* genes, and a spacer on different plasmids [82]. The spacer sequence is complementary to the targeted gene and can direct the Cas3 protein to the phages carrying the target gene (wild-type phages). This system was also used for the engineering of a *Vibrio cholerae* lytic phage while both donor DNA and CRISPR-Cas components were assembled in a single plasmid [83]. The efficiency of this system is much higher than that of BRED (~40%) [82] and can be used for genome engineering of mycobacteriophage to knock out the nonfunctional genes.

##### CRISPR–Cas9 (Type II)

The CRISPR/Cas9 system has been the first and most widely adopted CRISPR system for genetic engineering. Among bacteria, *Streptococcus thermophilus* CRISPR-Cas system type II-A has been applied to first insert point mutations, small and large DNA deletions, and gene replacements in virulent phage 2972 [84]. This approach has been applied to many other phages such as *Klebsiella* bacteriophage, *Bacillus* phages, T4, *Listeria* phages, etc. The components of this system should be delivered to an appropriate host cell by a single or multiple plasmids expressing (1) Cas9 protein; a nonspecific endonuclease, (2) CRISPR RNA (crRNA); a 17–20 nucleotide sequence complementary to the target DNA, (3) transactivating crRNA (tracrRNA); a binding scaffold for the Cas nuclease and (4) donor template DNA [85]. The crRNA and tracrRNA (guide RNA), which could be expressed in a single fusion RNA [86], are useful for directing the Cas nuclease to the specific DNA locus on the phage genome, where it makes a double-strand break during phage infection. The break will be repaired by recombination with the donor to generate mutants of interest. This system can be used for genome engineering and removing the undesirable genes from mycobacteriophages in future research.

##### CRISPR–Cas10 (Type III)

This system was used for engineering virulent staphylococcal phages where the donor DNA was cloned into the same plasmid expressing crRNA [87]. The infection of the host (*S. epidermidis*) with staphylococcal phages could result in Cas10-Csm cleavage of the wild-type phage genome and stimulate homology-directed repair using the donor region in the plasmid as a repair template with 100% efficiency [87]. Recently this system has been applied to gene editing (knocked-in/out) of the *M. tuberculosis* genome [88], highlighting its potential for mycobacteriophage engineering.

CRISPR-Cas systems are an important and very successful tool for the modification of phage genomes and will likely be adapted to additional bacterial species such as mycobacteria and mycobacteriophages in the near future. It will also allow for the targeting of toxic genes because short homology arms (50–150 bp) are sufficient to enrich the modified phages. However, this approach requires an active endogenous or heterologous CRISPR-Cas system in the phage propagation host, the availability of plasmid systems to carry different components of the systems, and a host that is relatively easy to transform. Furthermore, the overall procedure is time-consuming, and multiple editing steps have to be performed sequentially to isolate mutant phages. CRISPR-Cas systems are currently limited to a few bacterial hosts and phages, and there has been no published report for optimizing this system for mycobacteria and their phages.

#### 7.2.3. Rebooting Phages Using Assembled Phage Genomic DNA

Assembling well-defined phage cocktails for PT using natural phages is a very time-consuming and expensive process. Phages are selective for particular bacterial strains based on their binding to the host cell’s surface receptors. In addition, reliance on receptor recognition for infectivity implies that resistance against a phage can occur naturally through host receptor mutations. On the other hand, increasing the host-range of phages could increase the potential application of PT to treat the infection caused by different strains of pathogenic species. For example, one designed phage cocktail was only able to kill strain GD01 of *M. abscessus* subsp. *massiliense*, but not others (not a generalizable treatment) potentially due to the host-range barrier [25]. Phage engineering approaches can be used to generate multiple unrelated phages that collectively target a range of receptors, which may facilitate the creation of next-generation antimicrobials that decrease the chance of resistance development. Ando et al. were able to remove the host-range barriers across the genus for the T7 by swapping tail fibers through a phage engineering approach [89]. Different fragments of the phage genome can be synthesized or generated by PCR. A fragment encoding phage tail that has a narrow host-range feature can be replaced with a new fragment encoding phage tail (obtained from a broad host-range phage) conferring broad host-range at the species or genus level. Yeast artificial chromosomes (YAC) can be used to assemble the phage fragments in a yeast host. Then the assembled phage in a YAC vector can be transformed into appropriate host cells to generate mature phages with new tails conferring a broad host-range (Figure 4) [89]. *E. coli* can be used as a rebooting host for the phages that infect Gram-negative bacteria. Phages that infect Gram-positive bacteria cannot be rebooted in *E. coli*, but instead require a Gram-positive host, such as L-form cells, for transformation as rebooting compartments to overcome the very thick peptidoglycan (PG) layer of Gram-positive bacteria. L-form *Listeria* has been employed for *Listeria* phages a2s well as *Bacillus* and *Staphylococcus* phages (cross-genus rebooting system) [90]. Muddy, BPs, and ZoeJ have a narrow host-range, allowing them to infect only one strain of *M. abscessus* subsp. *massiliense* [25]. The isolation of phages with a broad host-range (infecting all members of the *M. abscessus* complex), will allow replacing the tail encoding genes of lytic phages (i.e., Muddy, TM4, D29, etc.), which are narrow host-range, with the tail encoding genes of broad host-range phages. This manipulation should result in new lytic phages able to infect and kill strains and subspecies of the *M. abscessus* complex (Figure 4) as reported for different *Listeria* serovars [91]. *M. smegmatis* L-forms can be used as a rebooting host [92] for future research for genetic engineering of mycobacteriophages.

### 7.3. Arming Mycobacteriophages

Modern genome-editing technologies have facilitated the development of engineered phages with increased efficacy by introducing new desirable properties, such as the elimination of lysogeny, changed host-range, and additional genes arming phages with secondary antimicrobials, etc. CRISPR-Cas systems can be used for genome engineering (see above) or can be delivered by phages and used to directly and specifically kill pathogens via targeting of the chromosome, the regulation of virulence gene expression, or degradation of plasmids carrying virulence genes or antibiotic resistance genes [93]. For example, CRISPR-Cas9 delivered by phagemids (plasmids packaged in phage capsids) was successfully programmed to kill and remove target virulence genes and antibiotic resistance genes of *S. aureus* [94] and *Galleria mellonella* [95]. Recently, an M13 phage carrying CRISPR-Cas9 was successfully used to deplete a targeted strain of *E. coli* in the gut, indicating the potential application of phage or phagemid vectors to deliver CRISPR Cas system targeting *M. abscessus* complex in the patient. This system could be programmed to different virulence and/or drug-resistant genes and selectively target the *M. abscessus* complex among lung commensal bacteria (Figure 5).

Type I CRISPR-Cas systems (Cas3 nuclease), can create a single-strand nick at the defined DNA sequence, followed by the processive exonucleolytic degradation of the targeted strand resulting in robust bacterial death regardless of the gene targeted and do not have apparent strain- or sequence-dependent activity [93]. The type I CRISPR-Cas system is widely distributed in prokaryotes but has not been detected in mycobacteria species [9]. As type I-B CRISPR-Cas systems are found in *Clostridium difficile* isolates, Selle et al., used a lethal genome-targeting CRISPR array delivered by a phage-targeting genome to kill the bacteria [96]. In this system, two independent strategies are employed to kill the target cell (1) harnessing the endogenous CRISPR-Cas system by crRNAs to cause irreparable genome damage and (2) replication, assembly, and lysis activity of phage. Although *M. abscessus* complex does not have CRISPR-Cas systems, it seems, however, that adding the CRISPR-Cas3 component to species-specific lytic phages (broad host-range phage) could increase the application of the lytic phage for PT against *M. abscessus* complex (Figure 6).

Although CRISPR-Cas has massive potential for the sequence-specific killing of pathogens, using such an approach in real-world environments needs further investigation. Targeting multiple bacterial species at the same time using CRISPR-Cas delivery is the primary challenge. Phages could be employed to deliver the CRISPR-Cas system; however, the host-ranges of most phages are narrow. Using engineered phages, the host-range of the phages can be expanded; however, this technology remains at a preliminary stage. Moreover, targeting host bacteria in spatially structured and complex microbial communities will provide an additional challenge that might reduce the encounter rates between phages and their host. Another issue is the evolution of resistance to CRISPR-Cas which is discussed in the next section.

## 8. Phage Resistant Mechanisms in Bacteria

Bacteria modify the structure of their surface phage receptors through mutations, and they can block the access of a phage to the receptor through the production of an excess of the extracellular matrix, or even by producing competitive inhibitors or blocking the injection of the genomic DNA of the phage (Figure 7) [97]. Endonucleases are also widely used by bacteria as a part of Restriction-Modification (R-M) systems, which can cleave phage DNA. Many bacteria are equipped with adaptive immunity through interfering CRISPR sequences which can be updated by the degradation of the injected phage DNA (Figure 7) [97]. Bacteria also are equipped with a two-component abortive infection system that can abort phage invasion. For example, the Rex system in phage lambda-lysogenic *E. coli* has two RexA and RexB proteins for protection against phages. After phage infection, a phage protein–DNA complex is produced as a replication or recombination intermediate which can activate RexA, resulting in the activation of RexB. RexB is an ion channel which allows the passage of monovalent cations through the bacterial inner membrane, destroying the membrane potential and killing the cell [97,98].

Prophage-mediated defense systems can protect bacteria from superinfection by the same or closely related phages. It has been reported that mycobacteriophage Sbash prophage colludes with its host (*M. smegmatis*) to confer highly specific defense (complete immunity) against infection by the unrelated mycobacteriophage Crossroads by a mechanism similar to the one proposed for the lambda RexAB system [99].

Genome analysis detected 1-8 prophage regions in the genome of different species of the *M. abscessus* complex encoding more than 20,000 viral and phage proteins [100]. In another study, 89 open reading frames (ORFs) were identified in the genome of a prophage (Araucaria) recovered from *M. abscessus* subsp. *bolletii* [100]. Prophage-mediated defense systems are predicted to be widespread in bacteria such as mycobacteria and mutually benefit the phage and the host mycobacteria, but the impact of this defense system on PT targeting mycobacterial disease, including the *M. abscessus* complex, has not yet been characterized. Some survivors were detected after challenging a larger culture of *M. abscessus* subsp. *massiliense* (strain GD01) with a phage cocktail, which were resistant to the BPs33ΔHTH-HRM10 and ZoeJΔ45 phages [25], indicating that some portion of the GD01 population was able to block phage infection. Further study is needed to deeply characterize the genome of those survivors (presence of prophages) and their surface receptors (receptor changes through mutations) for phages BPs and ZoeJ to understand how phage-resistant bacteria can evolve quickly. This is a major concern for PT.

Phage-inducible chromosomal islands (PICIs) are phage parasites that were detected in Gram-positive bacteria (*S. aureus*) and have the capacity to interfere with the reproduction of certain phages at the late stage of phage gene transcription [101]. Further studies are needed to characterize the presence of PICIs in the genome of mycobacteria and address their implication and potential negative impact PT. Multicopy phage-resistance (*mpr*) genes of *M. smegmatis* encoded a protein that confers resistance to mycobacteriophages L5 and D29 by changing the structure of the cell wall or membrane, which resulted in phage DNA injection [102].

## 9. Phage Mechanisms to Escape the Bacterial Antiphage System

In contrast to the various known antiphage systems of bacteria, the counteracting mechanisms of phages are poorly understood. Phages evolve to improve their binding to a new receptor while losing the ability to bind to another previously recognized receptor. For example, phage λ improved its binding to the different receptor on different host (*E. coli*) genotypes [103]. Moreover, the different enzymes encoded by phages, such as endosialidase, hyaluronanlyase, exopolysaccharide degrading enzyme, and alginase, grant the phage access to its receptors, which are covered by a surface component like a capsule or another exopolysaccharide compound [104]. A reduction in the number of recognition sites in a viral genome increases the probability of overcoming the defense provided by an R-M system. For example, phage λ reduces the number of EcoRI recognition sites in its genome through mutation [105]. Some viruses encode their methyltransferase (an enzyme that modifies the host DNA), to protect their genome from host restriction enzymes by adding a methyl group to phage genome [106]. The inactivation of CRISPR-Cas loci through mutations or deletions in *cas* genes essential for target cleavage or by deleting targeting spacers could result in the evolution of resistant phages. Some phages carry anti-CRISPR genes encoding anti-CRISPR proteins (Acr) which can interfere with and antagonize different CRISPR-Cas immune systems such as type I (I-E, I-F, I-D, I-C), type II (II-A, II-C), type III (III-B), and type V in bacteria [107]. For example, AcrIIA2 and AcrIIA4 inhibit the function of extensively used *S. pyogenes* Cas9 (spCas9), both in an in vitro bacterial test system and in a human cell-based genome editing assay, which could be a new challenge in PT [108]. Some phages, such as the ICP1 phage of *V. cholerae*, also encode their own CRISPR/Cas adaptive response to evade host innate immunity [109]. Moreover, it has been shown that CRISPR-Cas9 pressure could result in the evolution of the phage genome and generate a new mutant phage able to escape the CRISPR-Cas system, indicating that CRISPR-Cas might be a double-edged sword [110].

## 10. Limitations and Challenges

Due to emerging drug resistance in bacteria, bacteriophages are proposed as a new class of antibacterial, a serious alternative to antibiotics. Despite the recent advance in PT of *M. abscessus* infection [25], some important uncertainties and challenges could still hinder the development of modern PT. The fundamentals of phage pharmacokinetics in animals and humans are different from those of chemical drugs, as phages are a self-replicating elements of microbial communities within the body, with characteristic responses of the body to virions. Furthermore, the factors that determine phages’ tendency to successfully penetrate, circulate, and finally clear the body are not well known, mostly due to the extraordinary diversity of bacteriophages [111]. Obtaining regulatory approval for the therapeutic applications of phage cocktails can also be challenging because of the significant diversity of phages in terms of structure, life cycle, and genome organization and the potential interaction of phages with themselves and the host in the human body. Rapid and massive bacterial lysis by lytic phages could result in the subsequent release of bacteria cell wall components (e.g., lipopolysaccharides), which can induce adverse immune responses in the human host [112]. New genetic tools allow us to easily generate engineered lytic phages for therapeutic purposes as reported before for *M. abscessus* subsp. *massiliense* [25]; however, the effects of releasing such genetically engineered lytic phages, during and after the treatment, into the environment is not clear, particularly when the engineered lytic phages become available for public use. The massive release of genetically engineered phages into the environment may dangerously influence bacterial community dynamics, genome evolution, and ecosystem biogeochemistry.

## 11. Conclusions

Phage therapy is a promising alternative to combat superbugs, such as members of the *M. abscessus* complex; however, we still need to improve the efficiency of the phages by increasing their host-range and their lytic activity, using available cutting-edge genetic engineering tools. BRED has been applied to genome editing of mycobacteriophages before and could be considered as short-term solution for the manipulation of the mycobacteriophages; however, editing through CRISPR, rebooting and arming mycobacteriophages should be considered as a long-term solution to increase the application of phage therapy against the *M. abscessus* complex. Only a few available phages are able to effectively infect some strains of the *M. abscessus* complex, as most of them recovered on *M. smegmatis* as a host. Using the *M. abscessus* complex as a host will improve the availability of the new species-specific phages, which is critical for future phage therapy against the *M. abscessus* complex.

## Figures and Tables

**Figure 1 microorganisms-09-00596-f001:**
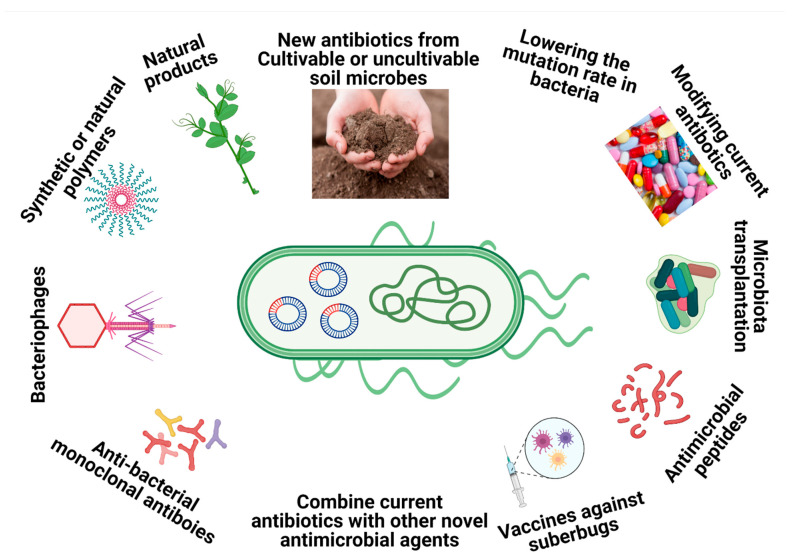
Overview of the main alternative strategies to combat superbugs.

**Figure 2 microorganisms-09-00596-f002:**
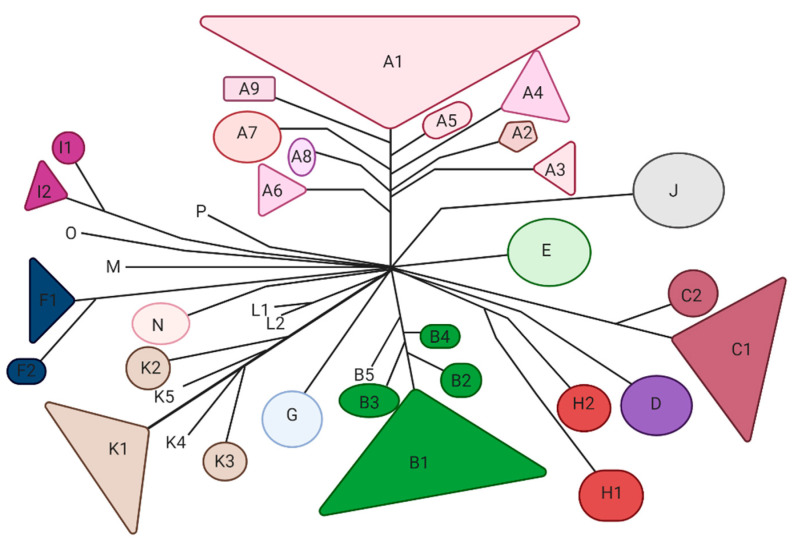
Overview of the main organizational clusters of mycobacteriophages. The clustering is based on the gene content of the sequenced mycobacteriophages reported previously [40,41].

**Figure 3 microorganisms-09-00596-f003:**
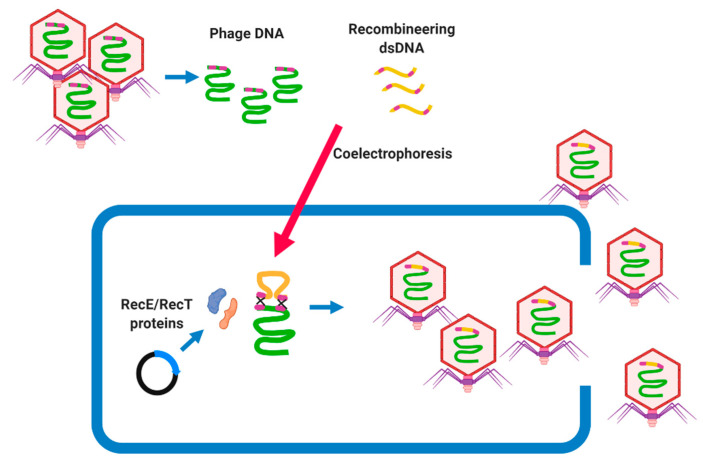
Overview of Bacteriophage Recombineering of Electroporated DNA (BRED) approach, which is popular for the genetic engineering of mycobacteriophages. The extracted phage genomic DNA and recombineering dsDNA (synthesized by PCR) is coelectroporated to the host cell carrying a plasmid encoding homologous recombination such as the RecE/RecT-like proteins. These proteins accelerate the homologous recombination between the phage DNA and recombineering dsDNA, which could result in a generation of phage mutants carrying the desirable trait.

**Figure 4 microorganisms-09-00596-f004:**
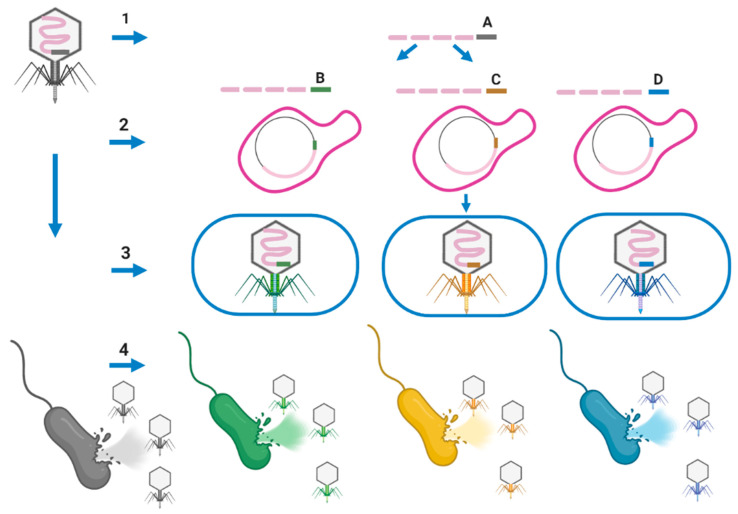
A useful model to improve the host-range of a lytic phage (i.e., Muddy) for *M. abscessus* complex. (1) Different fragments of the phage genome can be synthesized or generated by PCR. Each fragment should have over 30 bp homology with the adjacent fragment allowing them to assemble later in yeast (*Saccharomyces cerevisiae*). The first and last fragments of the phage genome should have arms that have homology with a yeast artificial chromosome (YAC) fragment. These arms can be added by PCR to those phage fragments. The corresponding fragment encoding tail of the original phage (fragment **A**) which confers a narrow host-range can be replaced by adding a fragment encoding new tail with a broad host-range (i.e., species-specific phage or genus-specific phage; **B**–**D**). (2) Then phage fragments consisting of a new tail fragment with broad host-range and YAC should be transferred to yeast for genome assembly of phages. The phage fragments will be recombined to form a complete phage genome in YAC using overlapping fragments with the new tail (broad host-range). (3) The assembled phage genome will be extracted from yeast and transferred into an appropriate rebooting host such as *M. smegmatis* to generate mature phages. (4) The engineered phage-targeting different strains or subspecies of the *M. abscessus* complex will be generated.

**Figure 5 microorganisms-09-00596-f005:**
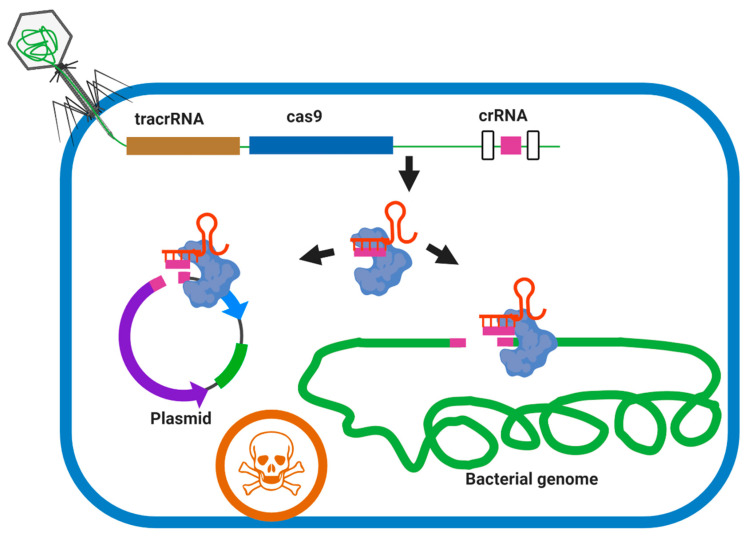
Sequence-specific killing of a target bacteria such as *M. abscessus* by a phagemid-delivered CRISPR-Cas 9 system. The broad host-range phage delivers a phagemid (carries the *S. pyogenes* tracrRNA, cas9 and a programmable CRISPR array sequence) to *M. abscessus* cells. Expression of cas9 and a self-targeting crRNA leads to chromosome cleavage and cell death. Virulence genes and drug-resistant genes can be targeted on both plasmid and genome by this approach in members of the *M. abscessus* complex.

**Figure 6 microorganisms-09-00596-f006:**
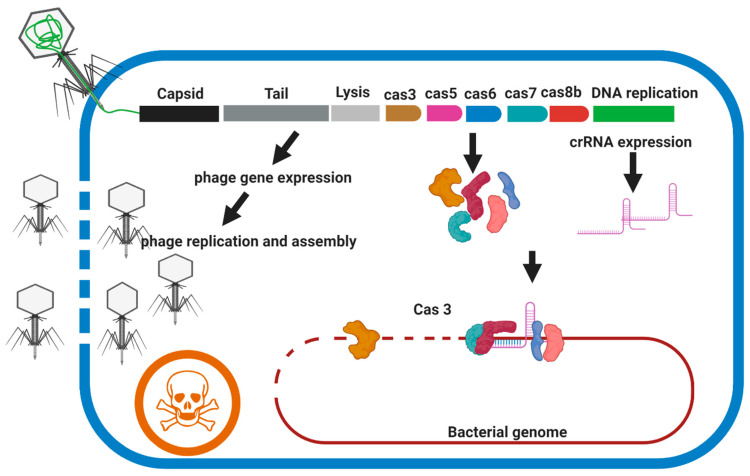
Overview of the acting mechanism of weaponized phage with CRISPR-Cas3. The genome of the candidate phage should be modified to encode a bacterial genome-targeting CRISPR array composed of a repeat-spacer-repeat meeting targeting conserved housekeeping genes of the *M. abscessus* complex. The genome-targeting CRISPR array and genes encoding CRISPR-Cas3 proteins are transduced into the bacterial cell during phage infection and are expressed concurrently with the lytic genes of the bacteriophage. Members of the *M. abscessus* complex are not expressing CRISPR-Cas 3 system endogenously. Cell death occurs by irreparable genome damage by Cas3 protein directed by the CRISPR RNA and cell lysis by the holin and endolysin expressed during lytic replication.

**Figure 7 microorganisms-09-00596-f007:**
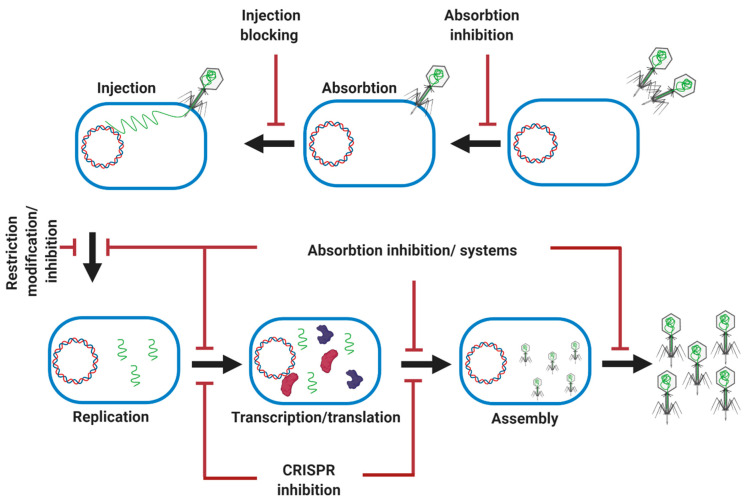
Summary of main microbial antiphage mechanisms. Microbes evolve different defense mechanisms to combat phage infection at different stages.

**Table 1 microorganisms-09-00596-t001:** Main applications of mycobacteriophages in a different setting.

Application	Purpose/Reported Phages
Diagnostic markers	Diagnosis of pulmonary tuberculosis (PhageTek MB kit) [43]
	Diagnosis of pulmonary tuberculosis (FASTPlaqueTB™) [44]
	Diagnosis of pulmonary tuberculosis (phage amplified assay: PhaB) [45]
Drug-resistant	Detection of isoniazid resistance (D29) [46] and Rifampin, isoniazid, ethambutol, streptomycin, and ciprofloxacin (D29) [47]
Genetic manipulation	Shuttle plasmids (L5, D29) [48], luciferase reporter phages (D29) [48], Recombineering (Che9c) [49]
Molecular typing	*M. tuberculosis* complex (GS4E) [50,51,52], *M. kansasii* (AX1, C3, KA3,6 and 8, D34A, D303-304, D345C) [53], *M. avium* (JF1-4, D302, and AN1-9) [54]
Therapeutic application	*M. tuberculosis* [55,56,57], *M. avium* [56,57], *M. ulcerans* [58], *M. abscessus* [25]

**Table 2 microorganisms-09-00596-t002:** Mycobacteriophages with lytic ability for different clinically important mycobacteria.

Host	Phage Cluster	Subcluster	Phage Name
*M. tuberculosis*	A	A1	Bxb1 and U2
		A2	L5 **a**, D29 **a**, Turbido
		A3	Bxz2 **a**, Microwolf, Rockstar, Vix
	B **b**	B1	Scoot17c
		B2	Qyrzyla
	G	-	Angel, Avrafan, BPs, Halo, Liefie, Bo4
	K	K1	Adephagia, CrimD, Jaws
		K2	TM4 **c**
		K3	Pixie
		K5	Fionnbharth
	Singleton	-	Dori
	F	F1	Ms6
	-	-	DS-6A, GR-21/T, My-327, BTCU-1, SWU1 **d**
*M. scrofulaceum*	D	D1	PBI1
	B	B1	PG2 **e**
	V	-	Wildcat **e**
*M. fortuitum, M. chelonae*	B	B4	Cooper
*M. avium*	K	K2	ZoeJ
*M. abscessus subsp. massiliense*	Singleton	-	Muddy (strain GD01)

a Bxz2, D29 and L5 have broad host-ranges and are effective on *M. tuberculosis*, BCG, *M. scrofulaceum, M. fortuitum, M. chelonae*, and some strains of both *M. ulcerans and M. avium*; b K2 group have broad host-ranges and are effective on *M. tuberculosis* and *M. avium*; c Plaque formation when plating large numbers of particles on *M. tuberculosis*; d These phages were not characterized; e Lytic for *M. fortuitum* and *M. chelonae*.

**Table 3 microorganisms-09-00596-t003:** Detail of the mycobacteriophages used to treat *M. abscessus* subsp. *massiliense* clinical case.

Phages	Source	Cluster/Subcluster	Length (kb)/Number of Coding Genes	Used in PT	Effectiveness Against GD01 Strain (PFU) a	GD01 Survivors	Effectiveness on Other Strains b
Muddy	Decomposed aubergine [62]	Singleton	48/72	Wild	Effective (<101–1010)	No	Ineffective
ZoeJ	Soil [63]	K/K2	57/92	Engineered (ZoeJΔ45)	Ineffective at low concentration (102–1010)	Yes	Ineffective
PBs	Soil [64]	G/G1	42/63	Engineered, mutant (BPsΔ33HTH-HRM10)	Ineffective at low concentration (108–1010)	Yes	Ineffective

## Data Availability

No new data were analyzed in this study. Data sharing is not applicable to this article.

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
