# Peer review of "Phage Therapy for Mycobacterium Abscessus and Strategies to Improve Outcomes"

_microorganisms, 2021, doi:10.3390/microorganisms9030596_

Round 1

Reviewer 1 Report

The authors reflect on therapy with a cocktail of three phages (one natural lytic phage and two engineered phages) against a severe case of disseminated M. abscessus subsp. massiliense infection. For this they have made a compilation of results where mycobacteriophages in a different setting have been used at different applications.

At the same time they are aware of limitations of phage therapy, such as the absence of lytic phages with a broad host-range against all strains and subspecies of M. abscessus complex and also risk of phage resistant bacteria over treatment.

I have looked at the article again and incorporated some improvement.

"The authors, in their review, analyze the treatment of of M. abscessu. They considere the resistance they have acquired against antibiotics. Therefore, in the review, it is considered phage therapy with a cocktail of three phages. But they have not highlighted the other strategies they show in their review. Perhaps they have opted to highlight only one strategy in that case their decision should be supported.

First, the authors note that the Members of M. abscessus are a new “antibiotic nightmare” as one of the most resistant organisms to chemotherapeutic agents. Regarding the bibliography in the introduction part it is satisfactory.

In the conclusion section, it could be included, which of the molecular biology techniques is the one that could be used at this time and which could be used in the long term.
About the included figures, they facilitate the compression of the main text."

Author Response

Comment 1:  "The authors, in their review, analyze the treatment of M. abscessus. They considered the resistance they have acquired against antibiotics. Therefore, in the review, it is considered phage therapy with a cocktail of three phages. But they have not highlighted the other strategies they show in their review. Perhaps they have opted to highlight only one strategy in that case their decision should be supported.

Response: Thanks for the comment. In this review, we only focused on phage therapy and its current status and future direction to combat superbugs such as M. abscessus. We listed other alternative that might be available against superbugs in the section “New Alternatives for Drug-resistant M. abscessus”. We added a figure (Figure 1) to highlight some of them.

Comment 2: “First, the authors note that the Members of M. abscessus are a new “antibiotic nightmare” as one of the most resistant organisms to chemotherapeutic agents. Regarding the bibliography in the introduction part, it is satisfactory.

Response: Thanks for the comment. Per reviewer 1’s comment, the members of M. abscessus complex are a new “antibiotic nightmare”. We highlighted this issue in “Introduction”.

Comment 3: In the conclusion section, it could be included, which of the molecular biology techniques is the one that could be used at this time and which could be used in the long term.

Response: Thanks for the comment. We added the following sentence to our conclusion following reviewer 1’s comment: “BRED applied to genome editing of mycobacteriophages before and could consider as a short-term solution for manipulating of the mycobacteriophages, however, editing through CRISPR, rebooting mycobacteriophages, and arming them should consider as long-term solutions to increase the application of phage therapy against M. abscessus complex”

Comment 4: About the included figures, they facilitate the compression of the main text."

Response: We added another figure per reviewer 2 suggestion as well to clarify more the alternative options to combat superbugs. 

Reviewer 2 Report

My suggestion would be that the paper is accepted without any further corrections. 

Author Response

We appreciate you for reviewing our manuscript. 

Reviewer 3 Report

The manuscript titled "Phage therapy for Mycobacterium abscessus and strategies to improve outcomes" by Abdolrazagh et al., is nicely compiled review. I have just one minor suggestions to provide before considering it for publication:

Introduction: It would be nice to see the illustrative presentation of all the alternative therapy currently in practice or under research to combat multidrug-resistant (MDR) bacteria such as M. abscessus.

Author Response

Comment 1: “Introduction: It would be nice to see the illustrative presentation of all the alternative therapy currently in practice or under research to combat multidrug-resistant (MDR) bacteria such as M. abscessus”

Response: Thanks for suggestion! We added another figure to the introduction section highlighting alternative therapy for superbugs such as M. abscessus.